# Analysis of Single Nucleotide Variants (SNVs) Induced by Exposure to PM_10_ in Lung Epithelial Cells Using Whole Genome Sequencing

**DOI:** 10.3390/ijerph18031046

**Published:** 2021-01-25

**Authors:** Se Jin Park, Gwan Woo Ku, Su Yel Lee, Daeun Kang, Wan Jin Hwang, In Beom Jeong, Sun Jung Kwon, Jaeku Kang, Ji Woong Son

**Affiliations:** 1Department of Internal Medicine, Konyang University Hospital, Daejeon 35365, Korea; fokidd@kyuh.ac.kr (S.J.P.); kde1324@naver.com (D.K.); sharpion@kyuh.ac.kr (I.B.J.); sjoongkwon@kyuh.ac.kr (S.J.K.); 2Department of Cardiothoracic Surgery, Konyang University Hospital, Daejeon 35365, Korea; kugwanwoo@kyuh.ac.kr; 3Myunggok Research Institute for Medical Science, Konyang University, Daejeon 35365, Korea; midanf@hanmail.net; 4Department of Thoracic and Cardiovascular Surgery, Seoul National University Bundang Hospital, Bundang 13620, Korea; hwj860523@hanmail.net; 5Department of Pharmacology, Myunggok Research Institute, College of Medicine, Konyang University, Daejeon 35365, Korea; jaeku@konyang.ac.kr

**Keywords:** particulate matter, lung cancer, single nucleotide variants

## Abstract

There are many epidemiological studies asserting that fine dust causes lung cancer, but the biological mechanism is not clear. This study was conducted to investigate the effect of PM_10_ (particulate matter less than 10 μm) on single nucleotide variants through whole genome sequencing in lung epithelial cancer cell lines (HCC-827, NCI-H358) that have been exposed to PM_10_. The two cell lines were exposed to PM_10_ for 15 days. We performed experimental and next generation sequencing analyses on experimental group that had been exposed to PM_10_ as well as an unexposed control group. After exposure to PM_10_, 3005 single nucleotide variants were newly identified in the NCI-H358 group, and 4402 mutations were identified in the HCC-827 group. We analyzed these single nucleotide variants with the Mutalisk program. We observed kataegis in chromosome 1 in NCI-H358 and chromosome 7 in HCC-827. In mutational signatures analysis, the COSMIC mutational signature 5 was highest in both HCC-827 and NCI-H358 groups, and each cosine similarity was 0.964 in HCC-827 and 0.979 in the NCI-H358 group. The etiology of COSMIC mutational signature 5 is unknown at present. Well-designed studies are needed to determine whether environmental factors, such as PM_10_, cause COSMIC mutational signature 5.

## 1. Introduction

Fine particles are particulate matter, a fine substance in a solid state. Fine particulate matter having a diameter smaller than 10 μm is called PM_10_. Most particulate matter (PM) comes from the combustion of fossil fuels. Recently, the concentration of fine particulate matter has been steadily increasing. The PM concentration in Korea has been measured as being greater than that of other countries, including the USA and Japan. In Korea, PM concentrations have been on the decline but are still high as compared to other countries [1]. PM enters humans through inhalation, and exposure to PM aggravates pulmonary inflammation because of its oxidative stress and direct toxic injury [2]. 

According to recent studies, exposure to PM has been found to affect the risk of cardiovascular disorders, cancer, neurologic disease, and many respiratory diseases, including chronic obstructive pulmonary disease, asthma and lung cancer [2,3,4]. Exposure to PM has been found to degrade lung function, and this affects many respiratory diseases. As the PM concentration increases, the number of patients with an acute exacerbation of asthma or chronic obstructive pulmonary disease (COPD) increases. 

As the PM exposure increases, the morbidity and mortality rate of COPD and asthma also escalate [5,6,7]. Other studies have reported an increased exacerbation frequency of idiopathic pulmonary fibrosis (IPF) because of decreased lung function [8]. Recent studies have shown that PM can also be a carcinogen [9,10], that PM concentration is associated with lung cancer development, and that there is positive correlation between the PM concentration and the incidence of lung cancer. Although several studies have investigated the epidemiological correlation between PM and lung cancer, the correlation between the molecular biological properties, including the mechanism of PM and the pathogenesis of lung cancer, is not clear. 

Mutations do not occur randomly. Depending upon their own preferred contexts, different mutational processes often result in different patterns of somatic mutations. These distinct patterns are called mutational signatures. Therefore, analyzing the pattern of the mutation can help us understand their causes. In addition to the mutational signature, kataegis, hypermutation confined to small genomic regions, is found in many types of cancer. Between transcribed and untranscribed gene strands, the efficiency of DNA damage and DNA maintenance processes can vary [11]. Thus, research about the mutational processes has a potential impact on the understanding and treatment of cancer.

This study was conducted to investigate the effect of PM_10_ on single nucleotide variants in lung cancer through whole genome sequencing of the lung epithelial cancer cell line (HCC-827, NCI-H358) exposed to PM. We used the Mutalisk program (National cancer center, Goyang-si, Korea) for analysis [12]. Additional information about Mutalisk is available on the following website: http://mutalisk.org/.

## 2. Method

### 2.1. Cell Culture and PM_10_ Treatment

In this study, two lung cancer cell lines were used: NCI-H358 and HCC-827. The former, NCI-H358, carries the EGFR mutation while the latter cancer cell line, HCC-827, carries K-ras mutation together with strong epithelial properties [13]. We used these two cell lines in order to find common results in lung cancer cells, regardless of the characteristics of each cell. We purchased the two cell lines from the Korea Cell Line Bank. 

We selected ERM^®^ CZ120 (Sigma-Aldrich, St. Louis, MO, USA) fine dust as our PM_10_ because it is easy to use and is used for Korean fine dust meter calibration, and the particle size is guaranteed. The ERM^®^ CZ120 fine dust is mainly composed of arsenic, cadmium, lead, nickel, and other additional ingredients.

Each of the two cell lines was dissolved in a 37 °C water for one minute in a frozen state and diluted in 10 mL of a medium (RPMI 1640 + 10% FBS + 1% antibiotics). After dilution, each medium was centrifuged for five minutes at 1500 rpm, and the supernatant was removed. After removing the supernatant, we diluted the medium again in 10 mL of culture medium and incubated it in 100Φ dishes. When the confluence of the cells in the 100Φ dishes reached about 80%, we used the following procedure to progress the cells to subculture. First, we removed 10 mL of medium. This was washed by adding 6 mL of PBS (phosphate-buffered saline) (Thermo Fisher Scientific, Waltham, MA, USA). The supernatant was removed, and then 0.25 trypsin-EDTA (Thermo Fisher Scientific, Waltham, MA, USA) 1 mL~2 mL was treated on the dish. After incubating this for five minutes in a CO_2_ incubator (Thermo Fisher Scientific, Waltham, MA, USA) at 37 °C, confirming the cells that had been separated through microscopy, the trypsin-EDTA was neutralized by mixing it with 5 mL of the culture medium. This was collected in a 10 mL tube, and the supernatant was removed after centrifuging. After adding 1 mL of the culture medium to the cell pellet, we calculated the number and viability of the cells by using the trypan blue method. Each cell was seeded in 3 × $10^{5}$ cells in 100Φ dishes. We added 10 mL of the culture medium. After that, each cell line was exposed to PM_10_ so that the final concentration of PM_10_ was 50 μg/mL. We subcultured this five times every three days and added the PM_10_ in the same manner as before. Thus, the cell lines were exposed to PM_10_ for a total of 15 days. The untreated cells were grown simultaneously, built under the same parameters with which the treated cell lines were harvested at same time points. 

### 2.2. Next Generation Sequencing (NGS) Analysis Workflow

#### 2.2.1. Library Construction

We performed NGS analysis in both groups. The workflow is as follows:

We extracted the genomic DNA of the lung epithelial cells (Wizard Genomic DNA purification Kit, Promega, A1120).

In order to glean a final library of 300~400 bp average insert size, each DNA fragment was sequenced in accordance with the Illumina Truseq DNA sample preparation guide. Using the covaris systems, 1 μg (TruSeq DNA PCR-free library) or 100 ng (TruSeq Nano DNA library) of genomic DNA was fragmented, which induced dsDNA fragments with 3′ or 5′ overhangs.

An end repair mix converted the dsDNA fragments with 3′ or 5′ overhangs into blunt ends. The 3′ to 5′ exonuclease detached the 3′ overhangs, and the 5′ overhangs were repleted with the polymerase. The appropriate library size was chosen following the end repair, incorporating different ratios of the sample purification beads.

To avoid the jointment of the dsDNA fragments during the adapter ligation reaction, a single “A” nucleotide was added to the 3′ ends of the blunted fragments.

Ligation of numerous indexing adapters to the blunt ends of the DNA fragments was carried out to lay out the groundwork for hybridization onto a flow cell. 

The enriched DNA library was amplified utilizing the PCR for sequencing. The PCR primer solution was operated for PCR, which allowed for the annealment of the ends of each adapters (Truseq Nano DNA library only).

#### 2.2.2. Clustering and Sequencing

The library needed to be loaded into a flow cell so as to capture dsDNA fragments situated on a lawn of surface-bound oligos, complementary to the library adapters, for the purpose of generating clusters of cells. By way of bridge amplification, each of the fragments underwent an amplification process, which resulted in distinct, clonal clusters. The completion of cluster generation was indicated by a signal that the templates were all set up for sequencing. Single bases were detected via Illumina SBS technology (Illumina, San Diego, CA, USA), as they were placed into DNA template strands, which operated based on the method utilizing proprietary reversible terminator.

#### 2.2.3. Generation of Raw Data 

Raw images and base calling were produced by the Illumina Platform, which embodied an integrated primary analysis software referred to as RTA (real time analysis). Illumina package bcl2fastq2-v2.20.0 was utilized to convert the BCL/cBCL (base calls) binary into FASTQ and to mismatch barcodes by setting the demultiplexing option to perfect match (value: 0).

#### 2.2.4. Read Mapping

The HiSeq sequencing system provided paired-end sequences, which are mapped to the human genome with Isaac aligner (iSAAC-04.18.11.09, Illumina, San Diego, CA, USA) where the reference sequence was the UCSC assembly hg19 (original GRCh37 from NCBI, Feb. 2009). Enabling the 32-mer seed-based search incorporated into the Isaac aligner, the foremost mapping candidates could be withdrawn and identified. Low quality 3′ end along with adapter sequences had been trimmed from the alignment in the process. 

A binary alignment output file (.bam) was also generated by the Isaac aligner, which was equipped with sorted and duplicate-marked data.

#### 2.2.5. SNVs, Small Indels Calling and Annotation

Identification of single-nucleotide variants (SNVs) and short insertions and deletions (Indels) were performed through read processing, in which Strelka was used to effectively filter out both low quality reads and PCR duplicates. Such a read realignment process was carried out by Strelka so as to bolster the accuracy level. In addition, the germline probability model-based variant genotyping was analyzed. A block-compressed genomic variant call format (gVCF) file was the result file that contained information on these variants. Variant-only VCF was generated by the extract variant, one of the utilities included in gvcftools package, as all non-variant blocks from the gVCF file were eliminated along with the filterification of low-quality and high-depth variants. Another program, referred to as SnpEff (v4.3t), came into play to annotate variants from the Variant-only VCF file, to which the in-house program and SnpEff were applied for its annotation, with extra databases including ESP6500, ClinVar and dbNSFP3.5.

#### 2.2.6. Structural Variant Calling

Manta (1.5.0) (Illumina, San Diego, CA, USA) was first enabled to recognize the specific input data and options through the configuration step before going into the procedural execution on a single node. For identification of structural variants and large indels, the entire genome sequencing structural variant calling analysis, with default options, was performed by Manta. Control-FREEC (11.5) was used to determine and pinpoint copy number variant with 10,000 window size and no additional options. It also incorporated GC-content bias to normalize read counts and XY for sex. The CNV types were classified based on genome ploidy value 2. Thus, values above two indicated gain, whereas values below 2 signified loss.

### 2.3. Data Analysis

As mentioned earlier, in this study, two cell lines, HCC-827 and NCI-H358, were used. SNVs were analyzed with the Mutalisk program. For analysis, we used definitions and statistical tools as defined in Mutalisk for localized hypermutation, mutational signatures, transcriptional strand bias, correlation coefficients of somatic mutations, epigenomic and transcriptional features (including GC contents), DNA replication timing, DNase hypersensitivity, and histone modification. The program input was a standard VCF file, obtained from whole genome sequencing of the differences between the cell lines exposed to PM_10_ and the control group.

## 3. Results

### 3.1. Overview of Whole Genome Sequencing Results

A total of 12,388 differences in the SNVs, six copy number variations (CNVs), and 763 structure variations (SVs) in NCI-H358, and 13,348 SNVs, 27 CNVs, and 891 SVs in the HCC-827, were accounted for when the PM10 exposed cell group and the control group in two cell lines were compared. The underlying purpose of this study is to investigate SNVs. In total, 3005 SNVs were newly investigated in NCI-H358 and 4402 SNVs were examined in HCC-837 in the PM10 group.

### 3.2. Localized Hypermutation

SNVs were closely examined in both treated and untreated cells. As shown in Figure 1, localized hypermutation or kataegis is visually laid out in rainfall plot format. To be more specific, in NCI-H358, chromosome 1 appeared to be the most localized with 516 mutations, followed by chromosome 5 with 413 mutations and chromosome 2 with 344 mutations. Meanwhile, chromosome 7 was shown to be the most localized with 519 mutations in HCC-827, which was followed by chromosome 14 with 445 mutations and chromosome 1 with 388 mutations.

### 3.3. Mutational Signature

COSMIC mutational signatures, packaged with 30 different mutation types ranging from COSMIC signature 1 to COSMIC signature 30, trigger various mutational processes to create unique combinations of mutation types [14]. Figure 2 shows the mutational signature decomposition results generated by Mutalisk, using the differences between cell lines exposed to PM_10_ and the control group, in both the NCI-H358 and the HCC-827 groups. The number of total mutations was 4402 in the NCI-H358 group and 3005 in the HCC-827 group. In the NCI-H358 group, the seven best decomposition models ranked by Bayesian information criterion (BIC) were the Catalogue of Somatic Mutation in Cancer (COSMIC) mutational signature 5, 18, 1, 13, 30, 2, and 10. On the other hand, in the HCC-827 group, the six best decomposition models were the COSMIC mutational signature 5, 16, 1, 20, 30, and 13. The COSMIC mutational signature 5 was the dominant type of mutational signatures in both NCI-H358 and HCC-827, and the cosine similarity score between the observed and decomposed distribution of mutations was equivalent to 0.979 in NCI-H358, and 0.964 in HCC-827.

### 3.4. Transcriptional Strand Bias

Figure 3 shows the SNV transcriptional strand bias in the differences between those cell lines exposed to PM_10_ and the control group of the two cell lines (NCI-H358, HCC-827). By utilizing the RefSeq Gene dataset, Mutalisk facilitates the annotation of the transcription strand information of somatic mutant pyrimidine bases (reference alleles of C or T) and calculates the enrichment for each mutation class [12]. With regard to the NCI-H358 group, the C > T mutations were highly enriched in the untranscribed regions, whereas in the transcribed regions, the C > A mutations were enriched within the NCI-H358 group. The C > T mutations demonstrated high enrichment in the untranscribed regions within the HCC-827 group as well. The ** *p*-value was <0.05, and the *p*-value was obtained by using a goodness of fit test. 

### 3.5. Genomic and Epigenomic Modification

Somatic mutations—each of which is mapped to genomic regions according to each feature, respectively—for each individual functional element (i.e., GC content, DNA replication timing, DNase I hypersensitivity, and histone modification) were fully examined utilizing Mutalisk as the program can recognize mutations in the vcf file that map to the coding region of the selected genome assembly. The GC-content, DNase I hypersensitivity and histone modification features are classified into low, intermediate, or high level, while the DNA replication timing is classified into three phases (early, intermediate, or late phase) [12]. 

Figure 4 shows the Pearson correlation coefficients between epigenetic modification density and the frequency of SNVs in NCI-H358 and HCC-827. The GC content has a positive correlation in the NCI-H358, and the H3K9me3 has a positive correlation in HCC-827. The other histone modifications have negative correlations. Figure 4 also shows the percentage of explained histone modification variance. The H3K27Ac, H3K9me3, and DNase HS were observed in the top third of the explained variance rate in both cell lines, and GC content, replication timing, and H2AZ were the same in the bottom third. 

## 4. Discussion

According to a recent study, exposure to PM causes genetic and epigenetic change. Some studies analyzed peripheral blood of individuals exposed to air pollution which shows that PM can cause DNA damage [15,16]. The process of gene expression includes DNA methylation, post-translational histone modification, histone variation, chromatin remodeling, and noncoding RNA [17]. Epigenetic factors such as exposure to PM, affect DNA methylation, modifying gene expression [18]. PM causes increased histone H4 acetylation in human airway epithelial cell lines (A549, BEAS-2B), which acts on IL-8 and COX-2 promoters to increase gene expression [19,20]. Exposure to PM also increases miRNA-222 and miRNA-21 [21]. All in all, PM can serve as a casual factor that can lead to a wide range of variations associated with genetic and epigenetic changes. Since no research analyzing the mutational signature of PM could be found, we performed the analysis for the purpose of this study.

In this study, we used the Mutalisk for COSMIC mutational signatures to identify somatic mutations. Some types of mutational signature etiologies are revealed. The COSMIC signature 5 has been found in all cancer types and in most cancer cells, but the etiology of the COSMIC signature 5 is unknown [22,23,24]. In the signature 5 mutations, FHIT loss was observed. This variation is related with factors such as air pollution (i.e., asbestos) and smoking [25]. Comparing the difference between the cell lines exposed to PM_10_ and the control group in NCI-H358 and HCC-827, the COSMIC mutational signature 5 was the highest in both NCI-H358 and HCC-827. Based on these results, we can conclude that COSMIC mutational signature 5 increases because of environmental factors, including PM. We can also safely conclude that PM_10_ acts as a carcinogen. 

Transcriptional strand bias, which has been identified in reporter gene assays and cancer genome sequences to reflect the activity of nucleotide excision repair (NER), along with somatic mutations for every functional element for genomic and epigenomic modification, were closely studied using Mutalisk. NER can be defined as a non-specific repair process that is activated when DNA distortions caused by mutagenic biochemical modifications are detected [26]. Our result from this study demonstrated that with regard to the NCI-H358 group, high enrichment of the C > T mutations in the untranscribed regions, and enrichment of the C > A mutations in the transcribed regions manifested. In the HCC-827 group, the high enrichment of the C > T mutations centered on the untranscribed regions. In addition, most of the functional elements associated with histone modification displayed negative correlations. From the changes spotted in functional elements, speculations can be made that some variations in the function or phenotype of the cells are likely to occur. Further research on the matter needs to be conducted with regard to the biological significance of these results. 

PM_10_ is not a single substance. It is composed of several substances. In this study, we used Sigma ERM-CZ 120 PM_10_. PM_10_ is composed of ionic components, metal components, and carbon components. Despite the fact that the two components of ERM^®^ CZ120, arsenic and nickel, are commonly referred to as causal factors of cancer [27,28], it is yet to be seen whether these specific heavy metal substances significantly contributed to our results. 

Most of the studies on COSMIC mutations have looked at the association of a single substances and COSMIC mutations [29,30]. On the other hand, in this study, PM_10_, which is composite material, was used, and this may produce different results from those previously known. As COSMIC mutational signature 5 was confirmed in comparison between the group exposed to PM_10_ and the control group, PM_10_ can be considered as a candidate for causing COSMIC mutational signature 5. Unlike our results, previous studies show G:C → T:A transversions. The reason for this difference might be caused by the difference of the target gene and used material. They investigated the TP53 mutation and used a single substance such as 3-nitrobenzanthrone, benzo[a]pyrene [31,32]. However, we investigated the whole genome sequence and used a complex compound (PM10). 

Concerning kategis, chromosome 1 was found to be most localized in NCI-H358 and chromosome 7 in HCC-827. 

The genes expressed predominantly on chromosome 1 varied for each individual type of lung cancer [33]. As for chromosome 7, which is widely reputed for its significant relations with the initiation and growth of lung cancer [34], EGFR and MET, for this chromosome, were found to play a major role with regard to lung cancer [35]. Some genes of chromosome 7 can also be held accountable, to a certain degree, for the survival period of patients with lung cancer [36].

There are several limitations in our study. First, we did not use normal cells in this study. Because there is an oncogene addiction to the driving mutation of a cancer cell, it could not be ruled out that COSMIC mutations were caused by its oncogene addiction. We used the cancer cell lines because normal cells were difficult to maintain in this study. Therefore, the number or type of mutations can be overestimated when using cancer cells as compared to normal cells. Second, this was not performed in one cell. This study did not proceed by the expansion of one cell. Some mutations may have occurred during the cell separation. Despite these limitations, many mutations were found in this study. Mutations occurring in both the HCC-827 and NCI-H358 cells were similarly observed and occupied a large portion of the total mutations. Considering these results, it can be concluded that PM affects SNVs. Third, the use of PM10 may be a limitation in this study because PM10 mainly settles in the upper airway. However, some studies describe the relationship between PM10 and lung disease [37,38], so this is unlikely to be a major limitation. Fourth, we used commercially available fine dust, which contains a lot of heavy metals. Therefore, the results may differ in some areas with different fine dust components.

To sum up the results, kataegis was observed in chromosome 1 of NCI-H358 and chromosome 7 of HCC-827. Signature 5 was the dominant type in both cell lines. A considerable number of mutations in transcribed strands of genes, epigenetics, and genetic elements have been found. We suggest that these can affect life-control phenomena.

## 5. Conclusions

In this study, PM10 exposure is illustrated as a causal factor that induces COSMIC mutational signature 5, which is found to have been present in many cancer types including lung cancer. This study needs further evaluation and confirmation through well-designed research.

## Figures and Tables

**Figure 1 ijerph-18-01046-f001:**
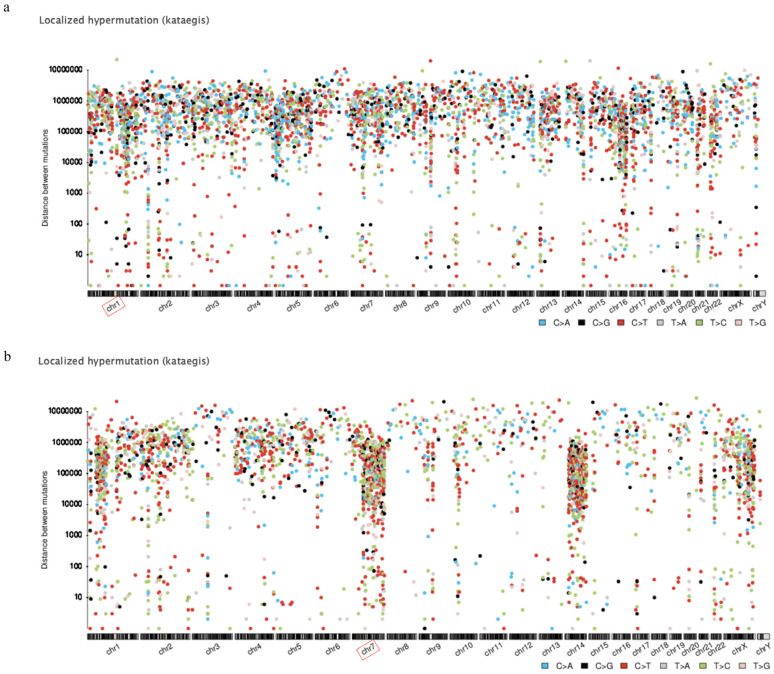
Localized hypermutation rainfall plot of the difference between the cell lines exposed to PM_10_ and control group in NCI-H358 (**a**), and HCC-827 (**b**). Each dot on the plot represents the physical genomic distance between each mutation. In NCI-H358, chromosome 1 appears to be most localized with 516 mutations, whereas chromosome 7 is most localized with 519 mutations in HCC-827. The figure was made using Mutalisk.

**Figure 2 ijerph-18-01046-f002:**
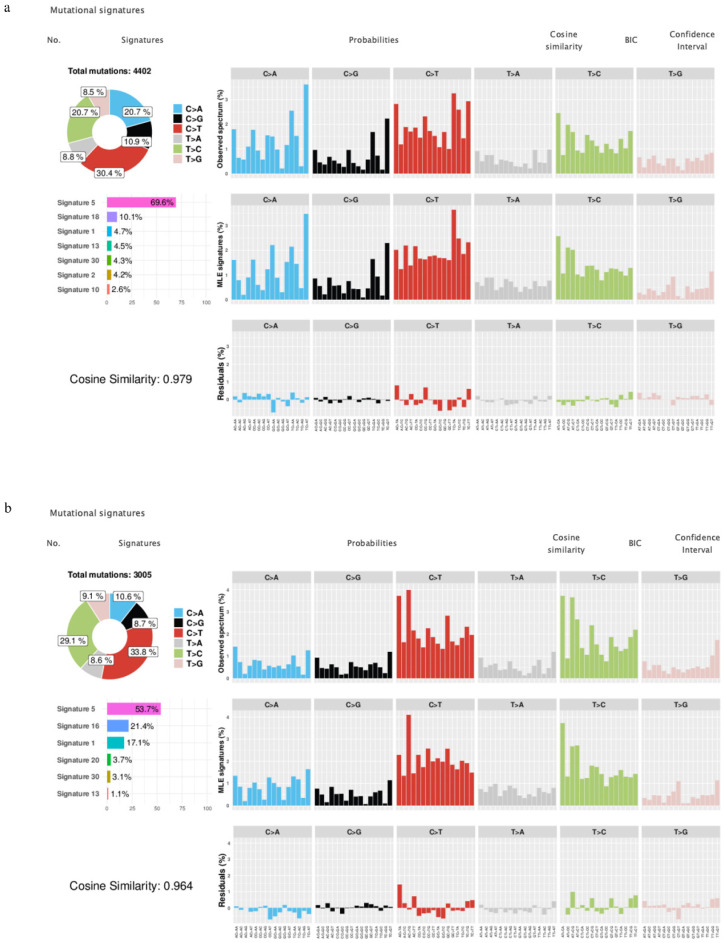
Mutational signatures: The best mutational signature decomposition model came from NCI-H358 (**a**) and HCC-827 (**b**). Signature 5 was the dominant type of mutational signatures in both the NCI-H358 and HCC-827 cell lines. The figure was made using Mutalisk.

**Figure 3 ijerph-18-01046-f003:**
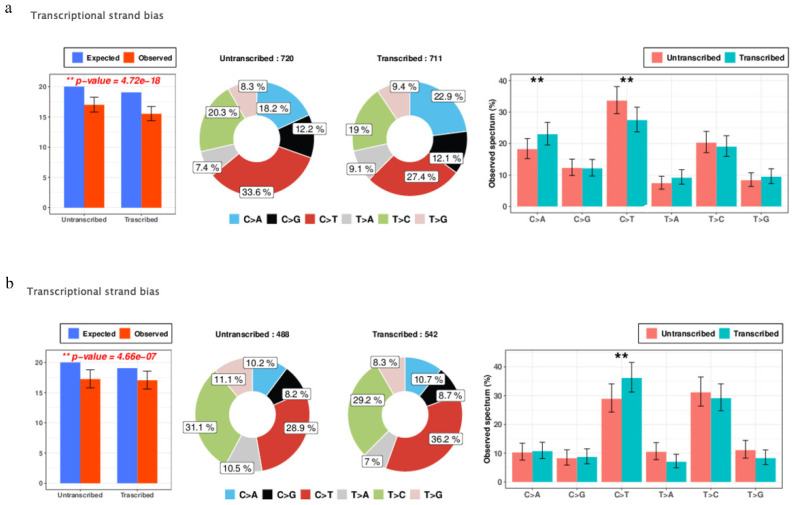
Transcriptional strand bias: Transcriptional strand bias analysis results in the difference between the cell lines exposed to PM_10_ and the control group in NCI-H358 (**a**), HCC-827 (**b**) (** *p*-value < 0.05). In the NCI-H358 group, high enrichment of the C > T mutations were demonstrated in the untranscribed regions and the C > A mutations were also enriched in the transcribed regions. In the other group (HCC-827), high enrichment of the C > T mutations were identified in the untranscribed regions. The *p*-value is obtained by a goodness of fit test. The figure was made using Mutalisk.

**Figure 4 ijerph-18-01046-f004:**
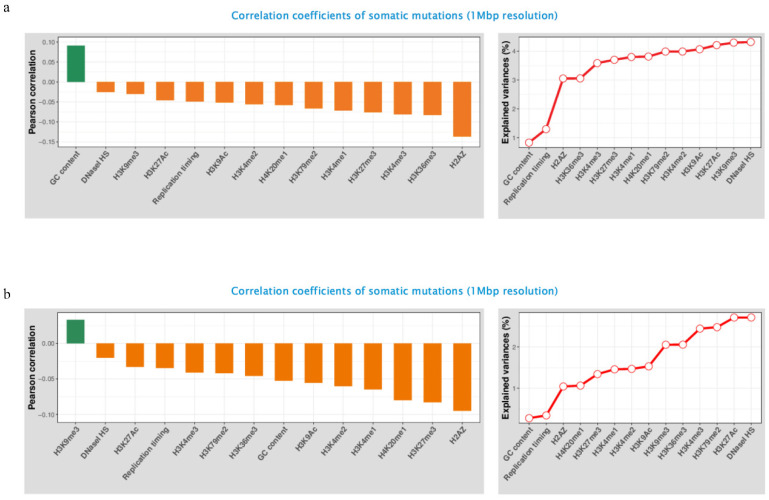
Pearson correlation coefficients and percentage of explained variance: Each Pearson correlation coefficient and the variance values show the computed differences between the cell lines exposed to PM10 and control group in NCI-H358 (**a**) and HCC-827 (**b**). Each Pearson correlation coefficient is computed between the read densities of 1-megabase bins and the mutation counts in the corresponding bins for each regulatory element. Bins without any mutation (=0) are excluded from the calculation, and mutations from both autosomal and sex chromosomes are included. The percentage of explained variance between the 14 elements and the mutations. The figure was made using Mutalisk.

## Data Availability

The data presented in this study are available on request from the corresponding author.

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
