# Peer review of "Analysis of Single Nucleotide Variants (SNVs) Induced by Exposure to PM10 in Lung Epithelial Cells Using Whole Genome Sequencing"

_ijerph, 2021, doi:10.3390/ijerph18031046_

Round 1

Reviewer 1 Report

In the current manuscript by Son et al., it is reported that exposure of cancer lung epithelial cells (HCC-827, NCI-H358) to particulate matter PM10 for 15 days, is associated with distinct patterns of single nucleotide variants (SNVs) in both cancer cells lines, that share a common COSMIC mutational signature 5. The present work is quite similar to a recent work published from the same group (in terms of NGS analysis) (‘’Particulate matter less than 10 μm (PM 10) activates cancer related genes in lung epithelial cells) relevant to the ability of PM10 to activate a variety of cancer related genes in lung epithelial cells.

The main concern in the present study is related to the lack of a normal lung epithelial cell line (as stated in the discussion by the authors), that could strengthen/further support current results, through comparison of the respective signatures in normal cells. Additionally, also correctly mentioned by the authors, in line 205: ‘’Because there is an oncogene addiction to the driving mutation of a cancer cell, it could not be ruled out that COSMIC mutations were caused by its oncogene addiction’’. In this context the observed results in the current study could not be fully attributed to the exposure of cells to PM10, and to this end title of the article should be changed. It is largely known that the genetic and epigenetic landscape of cancer cells is extremely different compared to that of healthy cells, characterized by a plethora of mutation signatures and altered epigenetic marks, that contribute to the onset/progression of the disease. Moreover, as COSMIC mutational signature 5 along with signature 1, are commonly present in all types of cancer cells, it is very difficult to assume that the observed mutational signature 5 in both cancer cell lines is a direct effect of PM incubation. This effect could be acceptable if observed in the case of normal lung cells treated with PM10.  

Another point that should be addressed, is the fact that beyond determination of a specific number of SNVs and mutational signatures, authors include in section 3.4 results in terms of Pearson correlation coefficients between the histone modification density and the frequency of SNVs, in HCC-827, NCI-H358 cells, but without stating at least in the discussion their biological significance in this type of cancer.

Generally, manuscript lacks of specific information especially in the introduction, while discussion does not provide any further comment(s) on the observed results (poorly analysed) as well as their direct connection (to a specific extent) following PM10 exposure. Especially lines 175-180, relevant to epigenetic changes/characteristics, should be completely rephrased and/or expanded, also stating the biological significance of miRNAs222/21, along additional general information included, relevant to PM10-induced alterations of epigenetic marks.

Author Response

In the current manuscript by Son et al., it is reported that exposure of cancer lung epithelial cells (HCC-827, NCI-H358) to particulate matter PM10 for 15 days, is associated with distinct patterns of single nucleotide variants (SNVs) in both cancer cells lines, that share a common COSMIC mutational signature 5. The present work is quite similar to a recent work published  from the same group (in terms of NGS analysis) (‘’Particulate matter less than 10 μm (PM 10) activates cancer related genes in lung epithelial cells) relevant to the ability of PM10 to activate a variety of cancer related genes in lung epithelial cells.

  • In the previous paper we investigated the differences in gene expression of lung epithelial cells. However, in this study we investigated the pattern of mutations through in silico In addition, the treatment period of PM10 is also different. In the previous study, the period was 7days and in this experiment, it was 15 days.

The main concern in the present study is related to the lack of a normal lung epithelial cell line (as stated in the discussion by the authors), that could strengthen/further support current results, through comparison of the respective signatures in normal cells. Additionally, also correctly mentioned by the authors, in line 205: ‘’Because there is an oncogene addiction to the driving mutation of a cancer cell, it could not be ruled out that COSMIC mutations were caused by its oncogene addiction’’. In this context the observed results in the current study could not be fully attributed to the exposure of cells to PM10, and to this end title of the article should be changed .

  • The HCC-827 cells are cancer cells with EGFR mutation, and the NCI-H358 cells are cancer cells with K-ras mutation. Because it showed the same results (signature 5) despite these different characters, we thought these are more likely to be PM10-induced changes than the oncogenic addiction.

It is largely known that the genetic and epigenetic landscape of cancer cells is extremely different compared to that of healthy cells, characterized by a plethora of mutation signatures and altered epigenetic marks, that contribute to the onset/progression of the disease. Moreover, as COSMIC mutational signature 5 along with signature 1, are commonly present in all types of cancer cells, it is very difficult to assume that the observed mutational signature 5 in both cancer cell lines is a direct effect of PM incubation . This effect could be acceptable if observed in the case of normal lung cells treated with PM10. 

  • Since normal lung cells are difficult to survive for a long time, two types of lung cancer cells with different mutations were used. As mentioned above, the two cell lines showed the same results, so we think these results could be acceptable.

Another point that should be addressed, is the fact that beyond determination of a specific number of SNVs and mutational signatures, authors include in section 3.4 results in terms of Pearson correlation coefficients between the histone modification density and the frequency of SNVs, in HCC-827, NCI-H358 cells, but without stating at least in the discussion their biological significance in this type of cancer .

  • We look at somatic mutations for each functional element (i.e., GC content, DNA replication timing, DNase I hypersensitivity, and histone modification). Most of the functional elements related to histone modification showed negative correlations, but further research is needed on the biological significance of this.

However, since there have been changes in functional elements, we speculate that there will be some changes in the function or phenotype of the cells. And following your advice, we added this content to the Discussion section.

Generally, manuscript lacks of specific information especially in the introduction, while discussion does not provide any further comment(s) on the observed results (poorly analysed) as well as their direct connection (to a specific extent) following PM10 exposure. Especially lines 175-180 , relevant to epigenetic changes/characteristics, should be completely rephrased and/or expanded, also stating the biological significance of miRNAs222/21, along additional general information included, relevant to PM10-induced alterations of epigenetic marks.

  • We mentioned it to explain that PM can cause a variety of genetic and epigenetic changes and to show there is no study to analyze the mutational signature of PM. In the Discussion section, we wanted to make the purpose of our research once again accurate.
  • We admitted that the previous sentences are misleading, and we corrected them. Thank you for your helpful advices.

Reviewer 2 Report

Park et al introduced the paper “Analysis of Single Nucleotide Variants (SNVs) induced by exposure to PM10 in lung epithelial cells using whole-genome sequencing” where they try to identify the SNPs associated with PM10 exposure. They used whole-genome sequencing to address this issue using two cancer cell lines  HCC-827 and  NCI-H358. 

Introduction:

The introduction section is well-written and easy to follow,  it describes both the disease cause and the idea behind the paper. Plus, it gives a good motivation.

Methods:

  • Which aligner the authors used and what parameters? 
  • What genome 37/38?
  • How the authors identified the difference in SNVs between the PM10 exposed and the control (12,388 and 13,348 )? And what is the percentage of this difference SNVs compared to all identified SNVs? How many SNVs the authors identified?
  • “In the PM10 group, 3005 SNVs were newly identified in NCI-H358, and 4402 SNVs were 107 identified in HCC-827 “  Compared to what?

Results:

  • In figure 1 can you please highlight/give more explanation to show which mutations have been clustered and the distance that they have clustered through. 

Author Response

Park et al introduced the paper “Analysis of Single Nucleotide Variants (SNVs) induced by exposure to PM10 in lung epithelial cells using whole-genome sequencing” where they try to identify the SNPs associated with PM10 exposure. They used whole-genome sequencing to address this issue using two cancer cell lines  HCC-827 and  NCI-H358.

Introduction:

The introduction section is well-written and easy to follow,  it describes both the disease cause and the idea behind the paper. Plus, it gives a good motivation.

Methods:

Which aligner the authors used and what parameters?

  • Thank you for your constructive advices. We have completely modified the method section to contain your questions. We sincerely ask you to check the revised manuscript.

What genome 37/38?

  • Please tell us in detail what needs to be corrected. We could not find the part 'genome 37/38'.

How the authors identified the difference in SNVs between the PM10 exposed and the control (12,388 and 13,348 )? And what is the percentage of this difference SNVs compared to all identified SNVs? How many SNVs the authors identified?

  • The result was derived by calculating the difference in the number of SNVs, CNVs and CVs in the PM10-treated group and the non-treated group based on variant statistics of the whole genome sequencing.
  • We attached the results by word file.

“In the PM10 group, 3005 SNVs were newly identified in NCI-H358, and 4402 SNVs were 107 identified in HCC-827. “ Compared to what?

  • It means the change in the PM10 treatment group compared to the control group.

Results:

In figure 1 can you please highlight/give more explanation to show which mutations have been clustered and the distance that they have clustered through.

  • Unfortunately, there were some errors during data processing, so Figure 1 was changed. We apologize for this. In NCI-H358, chromosome 1 was most localized with 516 mutations. In HCC-827, chromosome 7 was most localized with 519 mutations.
  • We got the merged results for each dot in figure 1 and counted it manually. 

Reviewer 3 Report

Dear authors,

Congratulations on submitting your work for publication in a peer-reviewed journal.  I have read your manuscript and need you please o address the following points which ill benefit your article.

Introduction:

Line 46: "The mortality rate from COPD and asthma also correlates positively with PM exposure."  Please include a reference in support of this statement.

Line 64: please reference Mutalisk regional paper.

Methods:

1- Cell culture and PM treatment: please clearly describe the untreated group.  Clearly indicating whether the untreated cells were grown at the same time following the same parameters and whether harvested at same time points as the treated cell lines.

2- Experimental and NGS analysis workflow:  Please explain what you mean by "experimental" and NGS sequencing.  What does experimental refer to here?  (Line 93)

Please indicate protocol/ commercial kit used to gDNA extraction.  Illumina TruseqPCR Free is a kit that generates sequencing libraries not gDNA extraction.

Please indicate sequencing parameters to include: sequencing machine (NextSeq?Hiseq?other?), read depth?, single reads/paired end sequencing? read lengths?

Please indicate the basic bioinformatics pipeline used for alignment and variant calling, and reference all of Strelka (v2.9.10), Control-FREEC (v11.5), and Manta (v1.5.0).

3- Data analysis: lines 103 to 108 should be moved to results section as this does not describe methods.

Results.

The result section needs a a new subsection describing the differences between treated and untreated cell lines from WGS data.  This is to describe the number and type of variants identified. 

3.1- Localized hypermutations:  lines 116 to 118 need paraphrasing.  Each results section must be self explanatory.  Therefore in section 1 of results, please start by explaining you are looking at SNV between treated and untreated cells, then simply describe the number of SNV variants you detected in each cell line.  Second, explain you looked at mutation clusters (hypermutations) and indicate which program you used to do so.  Indicate any parameters/statistical evidence of hypermutations in the regions you have shown.  The readers needs to understand and see evidence of how you reached the findings/conclusion that hypermutations are obvserved in Chr2 and 5 in NCI-H358 cell lines and ChrX in HCC-827 cell line.

3.2- Mutational signatures: please include a sentence explaining to the readers that is COSMIC mutational signatures, and what is cosine similarity.  Include a reference/link for COSMIC mutational signatures. 

Alos, please change the legend of Figure 2, to explain the findings rather than the methods.  Methods in Figure 2 legend should be moved to methods sectio

3.3. Transcriptional strand bias:  Lines 144 to 148 need paraphrasing.

3.4. Genomic and epigenomic modification.  Please explain to the reader what this section aims to describe (e.g. next, we looked at histone modification density in relation to localisation of hypermutations).  Explain how you performed the analysis such as the software used to detect epigenomic/histone modifications, then describe the results referring to Figure 4.  It is not clear what is "variance values"? "upper elements"? 

Discussion

The authors must address a key point in their discussion: the composition of the PM10 includes arsenic which was previously shown to cause lung cancer with a distinct mutational signature (PMID:24128716).  The authors must discuss this in line with their finings.  The authors must also include the known effects of the other components of their PM10 on their carcinogenic effect.   

You did not discuss nor included the in the results sections the genes located within the hypermutation areas.Are any of such genes known to be linked to cancer for example.  Please add a section in your results/discussion section.

Please explain to the readers why you chose to look at transcriptional strand bias and how it is relevant to PM10 causing cancer?

Line 177: please correct sentence to "...such as exposure to PM, affect DNA methylation, which causes CHANGES IN gene expression."

Line 187-188: "In the signature 5 mutations, FHIT loss was observed. This change mainly occurs when genes are exposed to carcinogens such asPM [22]".  You refer to the work of Nelson. et. al. which demonstrates that loss of FHIT is observed in many dance types and that this is associated with COSMIC mutational signature 5.  However, their results do not state this is dependent on exposure to PM.  In fact, Nelson et. al. clearly say that there was an "association" and not a dependency.  Therefore please rephrase this to make it more accurate and indicate that PM they re referring to are asbestos and smoking. 

Conclusion:

Line 223: you stated that "PM can induce the incidence of cancer by causing a COSMIC mutational signature 5.  Please chang this to reflect the findings of this study and which is: "PM10 exposure induces COSMIC mutational signature 5, which was previously shown to be present in many cancer types including lung cancer". 

Best wishes.

Author Response

Dear authors,

Congratulations on submitting your work for publication in a peer-reviewed journal.  I have read your manuscript and need you please o address the following points which ill benefit your article.

Introduction:

Line 46: "The mortality rate from COPD and asthma also correlates positively with PM exposure."  Please include a reference in support of this statement.

  • Thanks for the advice. We added references of the sentence.

Line 64: please reference Mutalisk regional paper.

  • We added the reference.

Methods:

1- Cell culture and PM treatment: please clearly describe the untreated group.  Clearly indicating whether the untreated cells were grown at the same time following the same parameters and whether harvested at same time points as the treated cell lines.

  • The untreated cells were grown at the same time following the same parameters and harvested at same time points as the treated cell lines. We described it to the methods section according to your advice.

2- Experimental and NGS analysis workflow:  Please explain what you mean by "experimental" and NGS sequencing.  What does experimental refer to here?  (Line 93)

  • Sorry for confusing you using unnecessary words. We deleted the word.

Please indicate protocol/ commercial kit used to gDNA extraction.  Illumina TruseqPCR Free is a kit that generates sequencing libraries not gDNA extraction.

Please indicate sequencing parameters to include: sequencing machine (NextSeq?Hiseq?other?), read depth?, single reads/paired end sequencing? read lengths?

Please indicate the basic bioinformatics pipeline used for alignment and variant calling, and reference all of Strelka (v2.9.10), Control-FREEC (v11.5), and Manta (v1.5.0).

  • Thank you for your helpful advices. We have completely modified the method section to contain your questions. We sincerely ask you to check the revised manuscript.

3- Data analysis: lines 103 to 108 should be moved to results section as this does not describe methods.

  • We moved the content to the first part of the results section by creating a new paragraph.

Results.

The result section needs a a new subsection describing the differences between treated and untreated cell lines from WGS data.  This is to describe the number and type of variants identified.

  • As in the answer above, we made a new paragraph 1 Overview of whole genome sequencing results to explain the NGS results.

3.1- Localized hypermutations:  lines 116 to 118 need paraphrasing.  Each results section must be self explanatory.  Therefore in section 1 of results, please start by explaining you are looking at SNV between treated and untreated cells, then simply describe the number of SNV variants you detected in each cell line.  Second, explain you looked at mutation clusters (hypermutations) and indicate which program you used to do so.  Indicate any parameters/statistical evidence of hypermutations in the regions you have shown.  The readers needs to understand and see evidence of how you reached the findings/conclusion that hypermutations are obvserved in Chr2 and 5 in NCI-H358 cell lines and ChrX in HCC-827 cell line.

  • Unfortunately, there were some errors during data processing, so Figure 1 was changed. We completely edited the section and explained how we reached the findings. Thank you for constructive advices.
  • We got the merged results for each dot in figure 1 and counted it manually.

3.2- Mutational signatures: please include a sentence explaining to the readers that is COSMIC mutational signatures, and what is cosine similarity.  Include a reference/link for COSMIC mutational signatures.

  • The description of 'COSMIC mutational signatures' was in the Discussion section. As you advised, we moved it to this part. The cosine similarity score is a measure of similarity between the observed distribution of mutations and the decomposed distribution of mutations.

Alos, please change the legend of Figure 2, to explain the findings rather than the methods.  Methods in Figure 2 legend should be moved to methods sectio

  • We edited the legend of Figure 2.

3.3. Transcriptional strand bias:  Lines 144 to 148 need paraphrasing.

  • We supplemented and revised the contents in that part.

3.4. Genomic and epigenomic modification.  Please explain to the reader what this section aims to describe (e.g. next, we looked at histone modification density in relation to localisation of hypermutations).  Explain how you performed the analysis such as the software used to detect epigenomic/histone modifications, then describe the results referring to Figure 4.  It is not clear what is "variance values"? "upper elements"?

  • At the beginning of the paragraph, we added an introduction to what we want to describe in this section. And we also added an explanation about analysis of genomic and epigenomic modification.
  • According to the explanation provided by Mutalisk, explained variance means the following: To quantify the extent to which the genomic and epigenomic properties can explain the somatic mutation-rate variation, Mutalisk calculates the percentage of explained variance for each vcf file using a previously reported method. This is achieved by forward feature selection where we iteratively select the genomic and epigenomic features with the lowestAkaike information criterion after fitting each remaining feature and the mutation data by a generalized least-squares estimation method. The percentage of explained variance is obtained from the linear regression model of these features.
  • 'The upper element' refers to 'the upper third' when the elements are listed in correlation order. The word 'the upper element' could cause confusion, so we changed it.

Discussion

The authors must address a key point in their discussion: the composition of the PM10 includes arsenic which was previously shown to cause lung cancer with a distinct mutational signature (PMID:24128716).  The authors must discuss this in line with their finings.  The authors must also include the known effects of the other components of their PM10 on their carcinogenic effect.  

  • We added the carcinogenicity of Arsenic and Nickel to the Discussion section.

You did not discuss nor included the in the results sections the genes located within the hypermutation areas. Are any of such genes known to be linked to cancer for example.  Please add a section in your results/discussion section.

  • We added information on chromosomes 1 and 7 that are most localized in the Discussion section.

Please explain to the readers why you chose to look at transcriptional strand bias and how it is relevant to PM10 causing cancer?

  • Transcriptional strand bias has been explained in reporter gene assays and cancer genome sequences. It is accepted as reflecting the activity of nucleotide excision repair (NER). NER is a non-specific repair process that is activated by detecting bulky DNA distortion caused by mutagenic biochemical modifications.
  • We added the explanation and our conclusion to the results in the Discussion section.

Line 177: please correct sentence to "...such as exposure to PM, affect DNA methylation, which causes CHANGES IN gene expression."

  • We changed the sentence according to your advice.

Line 187-188: "In the signature 5 mutations, FHIT loss was observed. This change mainly occurs when genes are exposed to carcinogens such asPM [22]".  You refer to the work of Nelson. et. al. which demonstrates that loss of FHIT is observed in many dance types and that this is associated with COSMIC mutational signature 5.  However, their results do not state this is dependent on exposure to PM.  In fact, Nelson et. al. clearly say that there was an "association" and not a dependency.  Therefore please rephrase this to make it more accurate and indicate that PM they re referring to are asbestos and smoking.

  • We change the sentence to the following: This change is associated with air pollution (i.e., asbestos) and smoking.

Conclusion:

Line 223: you stated that "PM can induce the incidence of cancer by causing a COSMIC mutational signature 5.  Please chang this to reflect the findings of this study and which is: "PM10 exposure induces COSMIC mutational signature 5, which was previously shown to be present in many cancer types including lung cancer".

  • We changed the sentence according to your advice. Finally, thank you so much for your delicate advices. Your advices made our paper constructive.

Round 2

Reviewer 1 Report

The revised version of the present work has been significantly improved, as several aspects throughout the manuscript have been clarified/expanded, especially in the materials and methods section.

It was clear based on my previous first comment (Report 1) that the present study was quite similar to a recent work by the same group ONLY in terms of NGS analysis

In addition authors have addressed my previous comments relevant to the lack of normal lung cell lines in their experiments, as well as my comments on correlation between epigenetic changes and functional elements.

Minor comment: Please correct line 245 to: ….between epigenetic modification density and the frequency of SNVs in……

and

lines 265-266 to: Epigenetic factors such as exposure to PM, affect DNA methylation, modifying gene expression

Author Response

20 January 2021

Dear editors and reviewers

We wish to express our appreciation to the editor and reviewers for their insightful comments, which have helped us significantly improve the paper. We highlighted the new modifications in green and kept the yellow highlights that were marked in the previous revision. We attached the details at the end. We hope that the changes incorporated into the revised manuscript satisfactorily address the reviewers’ concerns, and that our manuscript is now considered suitable for publication in your journal.

We thank you for your consideration and look forward to hearing from you.

Sincerely,

Dr. Ji Woong Son

[ Recommendation of reviewer 1 ]

Minor comment: Please correct line 245 to: ….between epigenetic modification density and the frequency of SNVs in……

  • Following your advice, we corrected the sentence.

lines 265-266 to: Epigenetic factors such as exposure to PM, affect DNA methylation, modifying gene expression.

  • Following your advice, we corrected the sentence and thank you for detailed advices.

Reviewer 2 Report

Congratulation on your work!

Author Response

20 January 2021

Dear editors and reviewers

We wish to express our appreciation to the editor and reviewers for their insightful comments, which have helped us significantly improve the paper. We highlighted the new modifications in green and kept the yellow highlights that were marked in the previous revision. We attached the details at the end. We hope that the changes incorporated into the revised manuscript satisfactorily address the reviewers’ concerns, and that our manuscript is now considered suitable for publication in your journal.

We thank you for your consideration and look forward to hearing from you.

Sincerely,

Dr. Ji Woong Son

[ Recommendation of reviewer 2 ]

Congratulation on your work!

  • Your constructive advices helped us improve our manuscript. Thank you so much.